# Anomalous spin current anisotropy in a noncollinear antiferromagnet

Cuimei Cao[1,10], Shiwei Chen [2,3,10], Rui-Chun Xiao[4], Zengtai Zhu[5,6], Guoqiang Yu [5,6], Yangping Wang[1], Xuepeng Qiu [3], Liang Liu[7,8], Tieyang Zhao[8], Ding-Fu Shao [9]✉, Yang Xu [1]✉, Jingsheng Chen [8]✉ & Qingfeng Zhan [1]✉

Cubic materials host high crystal symmetry and hence are not expected to support anisotropy in transport phenomena. In contrast to this common expectation, here we report an anomalous anisotropy of spin current can emerge in the (001) film of $Mn_3Pt$, a noncollinear antiferromagnetic spin source with face-centered cubic structure. Such spin current anisotropy originates from the intertwined time reversal-odd ($\mathscr{T}$-odd) and time reversal-even ($\mathscr{T}$-even) spin Hall effects. Based on symmetry analyses and experimental characterizations of the current-induced spin torques in $Mn_3Pt$-based heterostructures, we find that the spin current generated by $Mn_3Pt$ (001) exhibits exotic dependences on the current direction for all the spin components, deviating from that in conventional cubic systems. We also demonstrate that such an anisotropic spin current can be used to realize low-power spintronic applications such as the efficient field-free switching of the perpendicular magnetizations.

The anisotropy of transport phenomena is determined by the group symmetry and hence emerges in materials with low symmetry[1–6]. The high-sy6mmetry cubic materials widely used in electronic devices are usually not expected to host a strong transport anisotropy. A typical example is the widely used cubic structured spin source materials such as Pt, which hosts the isotropic spin Hall effect (SHE) to generate an out-of-plane spin current and the associated spin-orbit torque (SOT) for the manipulation of the magnetizations in spintronic devices[7]. The performance of such a device is independent of the current direction since the spin Hall conductivity (SHC) $\sigma_{zx}^y$ (in the form of $\sigma_{ij}^p$, where $i$, $j$, and $p$ are the generated spin-current, the driven

charge-current, and spin polarization directions, respectively) is invariant under rotation transformations of the coordinate system[8,9]. It would be interesting from the fundamental point of view and desirable for spintronic applications to find a new mechanism for the emergence of the anisotropic spin current in cubic spin sources even with the high symmetry film directions.

Here we demonstrate that an anomalous anisotropy of the spin current can be generated in noncolinear antiferromagnetic $Mn_3Y$ (Y= Pt, Ir, or Rh, to be distinguished from $Mn_3X$, X= Sn, Ge, or Ga) family with face-centered cubic structure even for the high symmetric (001) film, due to the noncolinear antiferromagnetism and hence the intertwined time-reversal-even ($\mathscr{T}$ − even) and time-reversal-odd

[1]Key Laboratory of Polar Materials and Devices (MOE), School of Physics and Electronic Science, East China Normal University, Shanghai 200241, People's Republic of China. [2]School of Physics, Hubei University, Wuhan 430062, People's Republic of China. [3]Shanghai Key Laboratory of Special Artificial Microstructure Materials, School of Physics Science and Engineering, Tongji University, Shanghai 200092, People's Republic of China. [4]Institute of Physical Science and Information Technology, Anhui University, Hefei 230601, People's Republic of China. [5]Songshan Lake Materials Laboratory, Dongguan, Guangdong 523808, People's Republic of China. [6]Beijing National Laboratory for Condensed Matter, Physics Institute of Physics, Chinese Academy of Sciences, Beijing 100190, People's Republic of China. [7]Key Laboratory of Artificial Structures and Quantum Control (Ministry of Education), School of Physics and Astronomy, Shanghai Jiao Tong University, Shanghai 200240, China. [8]Department of Materials Science and Engineering, National University of Singapore, Singapore, Singapore. [9]Key Laboratory of Materials Physics, Institute of Solid State Physics, HFIPS, Chinese Academy of Sciences, Hefei 230031, People's Republic of China. [10]These authors contributed equally: Cuimei Cao, Shiwei Chen. ✉e-mail: dfshao@issp.ac.cn; yxu@phy.ecnu.edu.cn; msecj@nus.edu.sg; qfzhan@phy.ecnu.edu.cn

($\mathscr{T}$ − odd) parts of SHE. The noncollinear magnetic configuration of Mn₃Y allows not only conventional SHC $\sigma_{zx}^{y}$ but also unconventional $\sigma_{zx}^{x}$, $\sigma_{zx}^{z}$, and all of them exhibit exotic dependence on the current direction. Using Mn₃Pt as a representative example, we confirm the anisotropic $\sigma_{zx}^{p}$ by the measurements of the current-induced spin-torque ferromagnetic resonance (ST-FMR) and the anomalous Hall effect (AHE) loop shift of the ferromagnetic layers adjacent to the Mn₃Pt layer in a SOT device. We also show this spin current can realize efficient field-free switching of the perpendicular magnetizations in ferromagnets, and the switching performances can be optimized according to such an anomalous anisotropy.

## Results

### Symmetry analyses

Mn₃Y (Y = Pt, Ir or Rh) is a material family that crystallizes in a cubic Cu₃Au-type structure[10,11]. As depicted in Fig. 1, the Mn atoms form kagome-type lattice planes stacked along the [111] direction. In its paramagnetic phase at high temperatures (Fig. 1a), the preserved time-reversal symmetry ($\mathscr{T}$) only allows the $\mathscr{T}$-even SHE[12,13], and the space group $Pm\bar{3}m$ enforces the isotropic $\sigma_{zx}^{y,\text{even}}$, i.e., $\sigma_{zx}^{y,\text{even}}(\phi_E = 0°) = \sigma_{zx}^{y,\text{even}}(\phi_E \neq 0°)$, where $\phi_E$ is used to denote the in-plane current direction with respect to an in-plane reference

direction [100]. In an SOT device where a ferromagnetic layer with a perpendicular magnetization is deposited on the Mn₃Y (001) film (Fig. 1b), an out-of-plane $y$-polarized spin current independent of the in-plane charge current direction can be generated in Mn₃Y, which enters the top ferromagnetic layer and exert a damping-like torque $\sim \mathbf{m} \times (\mathbf{m} \times \mathbf{y})$ to switch the ferromagnetic magnetization[12–14]. Such a switching requires a high current density and an external assisting magnetic field for deterministic switching and hence is inefficient for realistic applications[15–23].

Below the Néel temperature ($T_N$), Mn₃Y is antiferromagnetic with a noncollinear "head-to-head" or "tail-to-tail" alignments of Mn moments in the kagome planes (Fig. 1c). Its magnetic space group $R\bar{3}m'$ allows not only the conventional $\sigma_{zx}^{y,\text{even}}$ but also the unconventional $\sigma_{zx}^{x,\text{even}}$ and $\sigma_{zx}^{z,\text{even}}$. These SHC are dependent on the current direction as:

$$\begin{aligned}
\sigma_{zx}^{x,\text{even}}(\phi_E) &\propto \cos 2\phi_E, \\
\sigma_{zx}^{y,\text{even}}(\phi_E) &\propto \sin 2\phi_E, \\
\sigma_{zx}^{z,\text{even}}(\phi_E) &\propto \sin \phi_E - \cos \phi_E .
\end{aligned} \tag{1}$$

Therefore, the magnitudes of the $\mathscr{T}$ − even SHC are the same for currents along the primary directions [100] ($\phi_E = 0°$) and [010]

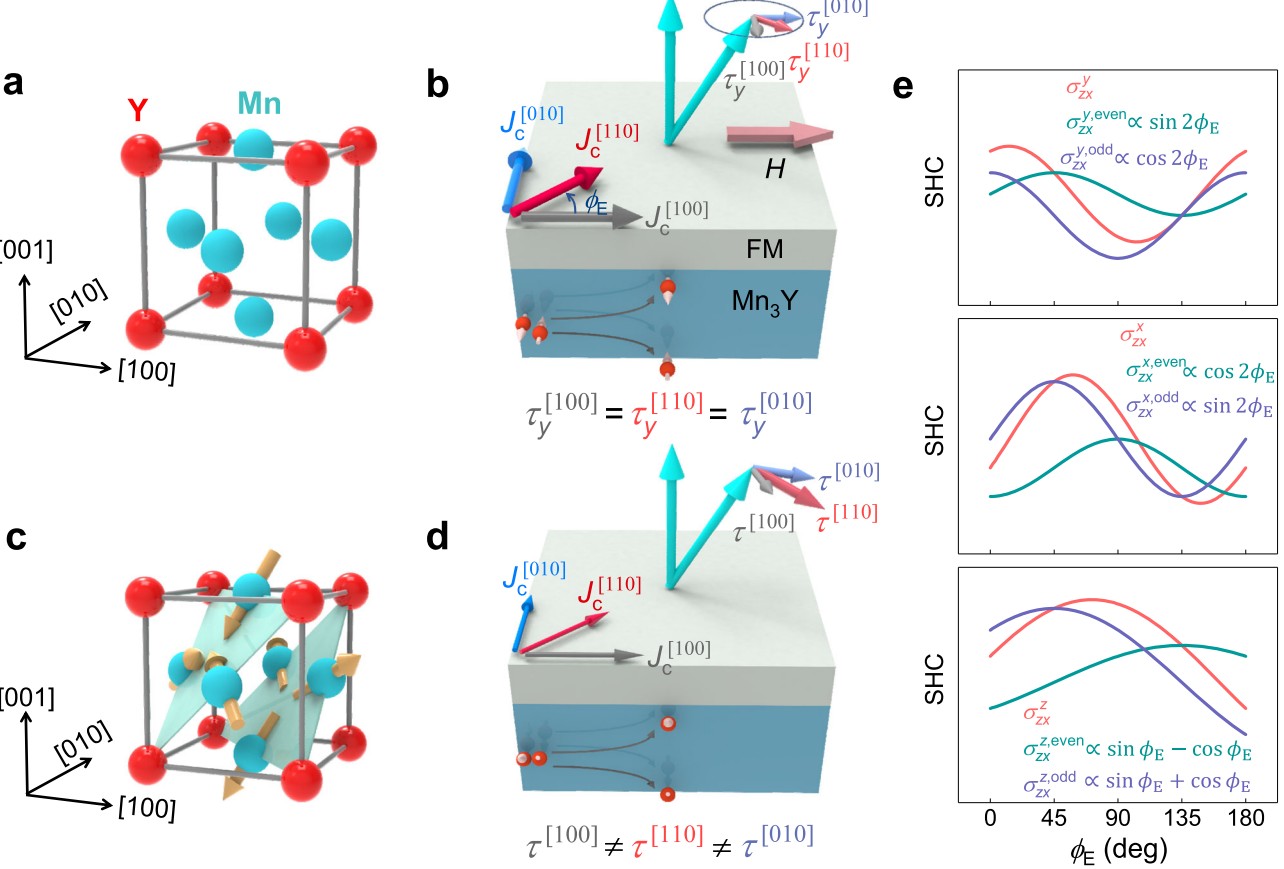

**Fig. 1 | Anisotropy of the spin current polarization and the associated SOT in spin source materials with a cubic structure. a** The crystal structure of cubic Mn₃Y (Y= Pt, Ir or Rh) in paramagnetic state. **b** A schematic of a SOT device using paramagnetic Mn₃Y as the spin source, where an isotropic and $y$-polarized spin current generated in the bottom Mn₃Y layer enters the adjacent ferromagnetic (FM) layer, exerting an isotropic SOT $\sim \mathbf{m} \times (\mathbf{m} \times \mathbf{y})$ on the perpendicular magnetization. $\phi_E$ is the angle between the current and the [100] direction of Mn₃Y. In this case, a sizable external magnetic field is required for a deterministic switching, and the charge current required is large. **c** The structure of cubic Mn₃Y with noncollinear antiferromagnetism, where the Mn moments form "head-to-head" or "tail-to-

tail" noncollinear alignments in (111) kagome planes. **d** A schematic of a SOT device using noncollinear antiferromagnetic Mn₃Y as the spin source, where an anisotropic spin current generated by Mn₃Y exerts the anisotropic SOT in the adjacent ferromagnetic layer. The presence of the $z$-polarization in the spin current and the associated unconventional SOC component $\sim \mathbf{m} \times (\mathbf{m} \times \mathbf{z})$ allows a field-free switching of perpendicular magnetization, which does not require a large charge current. **e** Theoretical $\phi_E$ dependence of the SHC $\sigma_{zx}^{p}$ ($p = y, x, z$) and its decomposition into the contributions from the $\mathscr{T}$-even and $\mathscr{T}$-odd SHE in Mn₃Y. The parameters used to plot (**e**) are shown in Supplementary Note 1.

($\phi_E = 90°$). On the other hand, the noncollinear antiferromagnetism breaks $\mathscr{T}$ symmetry and introduces non-spin-degenerate Fermi surfaces in Mn₃Y, which contribute to the $\mathscr{T}$-odd SHE (also termed as magnetic SHE, MSHE)[24] with the SHC:

$$\sigma_{zx}^{x,\text{odd}}(\phi_E) \propto \sin 2\phi_E,$$
$$\sigma_{zx}^{y,\text{odd}}(\phi_E) \propto \cos 2\phi_E, \qquad (2)$$
$$\sigma_{zx}^{z,\text{odd}}(\phi_E) \propto \sin \phi_E + \cos \phi_E.$$

These $\mathscr{T}$ − odd SHC for currents along the primary directions [100] ($\phi_E = 0°$) and [010] ($\phi_E = 90°$) also have the same magnitudes. However, the net SHC contributed by the intertwined $\mathscr{T}$-even and $\mathscr{T}$ − odd SHE has a more complicated anisotropy:

$$\sigma_{zx}^{x}(\phi_E) \propto \lambda_x \cos 2\phi_E + \mu_x \sin 2\phi_E,$$
$$\sigma_{zx}^{y}(\phi_E) \propto \lambda_y \sin 2\phi_E + \mu_y \cos 2\phi_E, \qquad (3)$$
$$\sigma_{zx}^{z}(\phi_E) \propto \lambda_z \cos \phi_E + \mu_z \sin \phi_E,$$

where $\lambda_i$ and $\mu_i$ are constants and can be used to estimate the relative strength of $\mathscr{T}$-even and $\mathscr{T}$-odd SHE. Figure 1e schematically show the $\phi_E$ dependence of $\sigma_{zx}$ and its decomposition into the $\mathscr{T}$-even and $\mathscr{T}$-odd components using arbitrary parameters (see discussion in Supplementary Note 1). Such anisotropy is anomalous for cubic systems, and particularly unexpected for the high symmetric (001) plane. Figure 1d illustrates the SOT exerted by the anisotropic spin current in Mn₃Y. The existence of an unconventional torque component $\sim \mathbf{m} \times (\mathbf{m} \times \mathbf{z})$ can directly change the effective damping and allows the efficient field-free switching of perpendicular magnetization with a small current[25–29]. One can further design the in-plane current direction for a maximum $z$-polarization in the out-of-plane spin current to optimize the performance of the SOT device.

## Characterization of the SOT associated with $\sigma_{ij}^{p}$ ($p = x, y, z$) by ST-FMR

Here, we use Mn₃Pt to demonstrate the anomalous anisotropy of spin current in Mn₃Y, which has a $T_N$ of ~475 K[10,11,30,31]. We first investigate the current-induced SOT in (001)-oriented Mn₃Pt/permalloy (Py) heterostructures by the ST-FMR technique. More details about the sample preparation and characterization are provided in Methods and Supplementary Note 2. The Mn₃Pt/Py heterostructure is patterned with microwave-compatible contacts whose orientation is varied to study the SOT anisotropy as a function of the azimuthal angle $\phi_E$, i.e., the angle between the microwave current $I_{rf}$ and the [100] direction of Mn₃Pt, as illustrated in Fig. 2a. Here the $x, y, z$ axes form local frames that change with the direction of $I_{rf}$, i.e., $I_{rf}$ is always along $x$. For each device with a fixed $\phi_E$, an in-plane external magnetic field $H_{ext}^{\parallel}$ is swept at an angle $\phi_H$ with respect to $I_{rf}$ (Fig. 2a). The ST-FMR signal is a rectified voltage $V_{mix}$, whose lineshape can be decomposed into a symmetric component $V_s$ and an antisymmetric component $V_a$ near the resonant condition[32], characterizing the in-plane SOT $\tau_{\parallel}$ and out-of-plane SOT $\tau_{\perp}$, respectively, allowing the full determination of the damping-like and field-like SOT from all possible spin polarizations (Supplementary Note 3)[33,34]. For the in-plane $\mathbf{m}$ of Py, $\tau_{\perp}$ may come from the damping-like SOT associated with $\sigma_{zx}^{z}$, while it also includes the field-like contribution from $\sigma_{zx}^{y}$, $\sigma_{zx}^{x}$, and the Oersted field.

In our Pt/Py reference sample, $V_{mix}$ is symmetric about $H_{ext}^{\parallel}$, since the spin current is generated solely from the conventional SHE and Oersted field (Supplementary Note 3). In Mn₃Pt, however, the measured $V_{mix}$ ($\phi_H = 10°$) does not overlap with $−V_{mix}$ ($\phi_H = 190°$), i.e., $V_{mix}$ does not exhibit a perfect inversion with the reversal of $H_{ext}^{\parallel}$, indicating the presence of unconventional SOT associated with the SHC other than $\sigma_{zx}^{y}$ (Supplementary Note 3)[1,35–40].

For further investigation, we measure the $\phi_H$ dependence of $V_a$ and $V_s$ at various $\phi_E$ (Supplementary Note 4). The $\phi_H$ dependence can

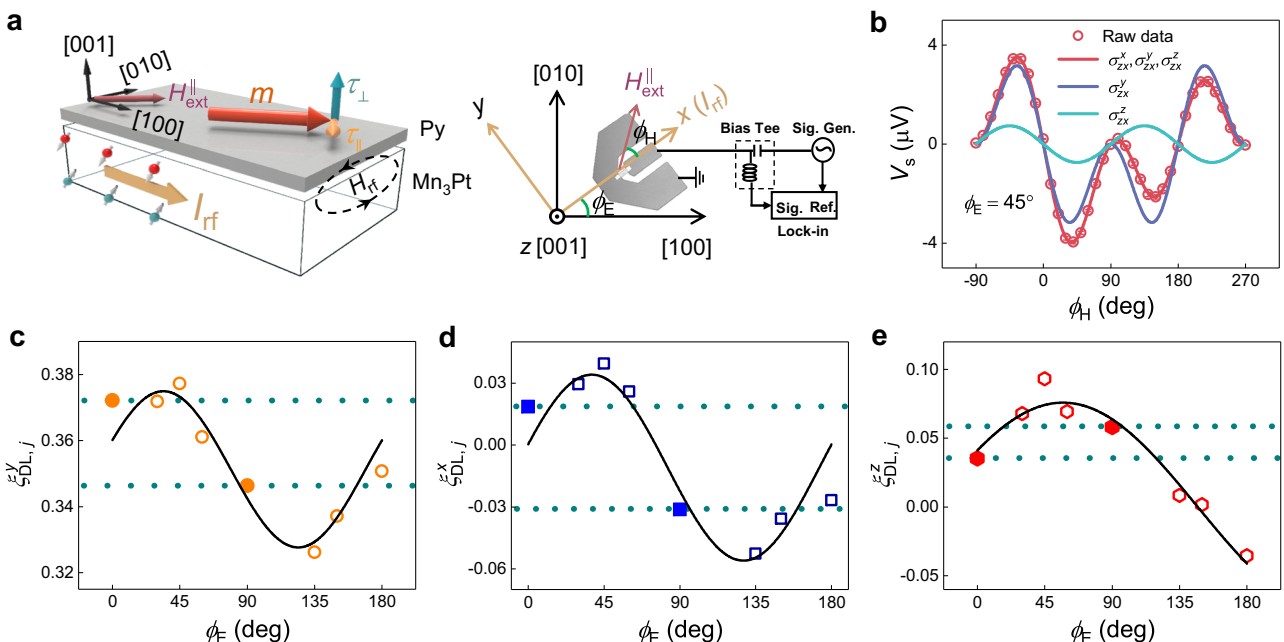

**Fig. 2 | Characterization of the SOT associated with $\sigma_{ij}^{p}$ ($p = x, y, z$) by ST-FMR in Mn3Pt/Py. a** (Left) The schematic of the Mn₃Pt/Py device. The magnetization **m** of the Py layer is set by an in-plane field and then subject to the in-plane SOT $\tau_{\parallel}$ and out-of-plane SOT $\tau_{\perp}$ associated with the spin current generated in Mn₃Pt. (Right) The schematic of the ST-FMR measurement setup. The electrical current $I_{rf}$ is injected with an angle $\phi_E$ relative to the [100] direction of Mn₃Pt. For each device with a fixed $\phi_E$, an in-plane external magnetic field $H_{ext}^{\parallel}$ is swept at an angle $\phi_H$ with respect to $I_{rf}$. **b** Representative $\phi_H$ dependence of $V_s$ at $\phi_E = 45°$. The decomposition into contributions from the spin polarizations $\sigma_{zx}^{p}$ can be derived from fittings (solid lines) to Eq. (5). **c–e** The $\phi_E$ dependence of the damping-like SOT efficiency per unit current density $\xi_{DL,j}^{p}$ associated with $\sigma_{ij}^{p}$. The solid lines are fitting lines using Eq. (3). The anisotropy between $\phi_E = 0$ and $\phi_E = 90°$ is highlighted by the enlarged symbols and dashed horizontal lines.

be fitted by[1,32,36,41]:

$$V_a = A_{FL}^y \cos\phi_H \sin 2\phi_H + A_{FL}^x \sin\phi_H \sin 2\phi_H + A_{DL}^z \sin 2\phi_H, \quad (4)$$

$$V_s = S_{DL}^y \cos\phi_H \sin 2\phi_H + S_{DL}^x \sin\phi_H \sin 2\phi_H + S_{FL}^z \sin 2\phi_H, \quad (5)$$

where $S_{DL}^y$, $S_{DL}^x$, and $A_{DL}^z$ are coefficients for the damping-like SOT associated with $\sigma_{zx}^y$, $\sigma_{zx}^x$, and $\sigma_{zx}^z$, respectively, while $A_{FL}^y$, $A_{FL}^x$, and $S_{FL}^z$ correspond to the field-like counterparts. Note that the contribution of the field-like SOT from the Oersted field is included in $A_{FL}^y$. We show in Fig. 2b how $V_s$ ($\phi_E = 45°$) is decomposed into the contributions from $\sigma_{zx}^p$ (see Supplementary Note 4 for $V_a$ and $V_s$ at $\phi_E = 0°$, $45°$, and $90°$). Important observations can be made: (i) Despite a subdominant contribution, the presence of nonzero $\sigma_{zx}^x$ and $\sigma_{zx}^z$ is evident, consistent with the asymmetric $V_{mix}(H_{ext}^{\|})$ mentioned above. (ii) Compared to the case of $\phi_E = 0°$, the $\sigma_{zx}^z$ contribution is enhanced for $\phi_E = 45°$ (Supplementary Note 4), consistent with the previous observation of a stronger (weaker) $\sigma_{zx}^z$ for $I_{rf}$ applied parallel (perpendicular) to the magnetic mirror plane of Mn$_3$Pt[38]. (iii) A notable discrepancy can be seen between the cases of $\phi_E = 0°$ and $\phi_E = 90°$, which is unexpected considering their equivalency in both the crystal and magnetic structures. This is different to previous reports in cubic systems where the anisotropic spin currents have never been observed in (001) films with the highest crystal symmetry[37,42].

To better visualize the SOT anisotropy, we show in Fig. 2c–e the $\phi_E$ dependence of the damping-like SOT efficiency per unit current density $\xi_{DL,j}^p$, which can be estimated by[32,39]:

$$\xi_{DL,j}^y = \frac{S_{DL}^y}{A_{FL}^y} \frac{e\mu_0 M_s t_{HM} t_{FM}}{\hbar} \left[1 + (M_{eff}/H_0)\right]^{\frac{1}{2}}, \quad (6)$$

$$\xi_{DL,j}^x = \frac{S_{DL}^x}{A_{FL}^y} \frac{e\mu_0 M_s t_{HM} t_{FM}}{\hbar} \left[1 + (M_{eff}/H_0)\right]^{\frac{1}{2}}, \quad (7)$$

$$\xi_{DL,j}^z = \frac{A_{DL}^z}{A_{FL}^y} \frac{e\mu_0 M_s t_{HM} t_{FM}}{\hbar}, \quad (8)$$

where $e$, $\hbar$, $t_{HM}$, $t_{FM}$, and $M_s$ represent the elementary charge, the reduced Planck constant, the thickness of the heavy metal (HM, Mn$_3$Pt) and ferromagnetic (FM, Py) layer, and the saturation magnetization of the FM layer, respectively. The effective magnetization $M_{eff}$ can be obtained from ST-FMR measurements performed at a sequence of microwave frequencies $f$ following the Kittel relation. Note that in our samples $A_{FL}^y$ is dominated by the Oersted field (Supplementary Note 5), making $A_{FL}^y$ a good measure of the current density in the Mn$_3$Pt layer. In principle, one can derive the relative magnitudes of $\sigma_{zx}^y$, $\sigma_{zx}^x$, and $\sigma_{zx}^z$ by fitting the $\phi_E$ dependence of $\xi_{DL,j}^p$ using Eq. (3). The fitting yields nonzero values of $\lambda_y = 0.04359$, $\mu_y = 0.01126$, $\lambda_x = 0.00328$, $\mu_x = 0.02038$, $\lambda_z = 0.02049$, and $\mu_z = 0.03176$. The values of $|\lambda_i/\mu_i|$ deviating from 1 is clear evidence of the intertwined $\mathscr{T}$-even and $\mathscr{T}$-odd SHE, resulting in a discrepancy between the signal at $\phi_E = 0°$ and $\phi_E = 90°$, manifested as the different values of $\xi_{DL,j}^i$ ($\phi_E = 0°$) and $\xi_{DL,j}^i$ ($\phi_E = 90°$). Such an anomalous anisotropy between the spin current polarization and the associated SOT along two orthogonal cubic directions is the main experimental finding of this work.

## AHE loop shift with a threshold current

Since the unconventional $z$-polarized spin current is important for low-power switching of high-density spintronic devices with perpendicular magnetizations, we build Mn$_3$Pt(5)/Ti(3)/CoFeB(1)/MgO(2)/SiO$_2$(2) heterostructures (numbers in parentheses indicate layer thickness in nanometers) with perpendicular magnetic anisotropy (PMA) to further quantify $\sigma_{zx}^z$ (Fig. 3a). The nonmagnetic interlayer Ti was used to

provide the PMA of CoFeB and magnetically decouple the Mn$_3$Pt and CoFeB layers[43], and hence it does not contribute to the magnetization switching (Supplementary Note 6). The SOT contribution from the Ti layer is negligible due to the extremely small spin Hall angle of Ti[43,44]. The PMA of this heterostructure is confirmed by the square hysteresis loop in the Hall resistance $R_{xy}$ as a function of the out-of-plane magnetic field $H_{ext}^z$ (Supplementary Note 7).

We first perform the AHE loop shift measurement on the PMA heterostructures[45]. As shown in Fig. 3b, the AHE loop under an out-of-plane magnetic field $H_{ext}^z$ and $I = 1$ mA along [110] (i.e., $\phi_E = 45°$) almost overlaps with the loop under $I = -1$ mA. However, as shown in Fig. 3c, when $I$ is increased to $\pm16$ mA, considerable AHE loop shift occurs, i.e., the center of the loop is shifted to positive (negative) field values for positive (negative) $I$. Such a shift is indicative of an effective field $H_{eff}^z$ acting in conjunction with the external $H_{ext}^z$ (see also Supplementary Note 8 and Note 9). Figure 3d depicts the $I$ dependence of $H_{eff}^z$ (see "Methods") for selected $\phi_E$ of $0°$, $45°$, and $90°$. Echoing the qualitatively different behavior between $I = \pm1$ and $\pm16$ mA, a clear threshold current effect is evident: $H_{eff}^z$ abruptly increases from near zero when $I$ is greater than a threshold value[45], characteristic of a damping-like $\tau_\perp$ originating from $\sigma_{zx}^z$[6,29,43,45].

To better visualize the anisotropy of $H_{eff}^z$ which characterizes the anisotropy of $\sigma_{zx}^z$ and the associated $\tau_\perp$, we plot in Fig. 3e the $\phi_E$ dependence of the $H_{eff}^z/J$, where $J$ is the applied in-plane current density. With increasing $\phi_E$, $H_{eff}^z/J$ first increases to a maximum for $\phi_E = 45°$, and then decreases and changes the sign around $\phi_E = 135°$. The $H_{eff}^z/J$ for $\phi_E = 0°$ is about a half of that for $\phi_E = 90°$. This behavior is consistent with the $\phi_E$ dependence of $\xi_{DL,j}^z$ measured by ST-FMR (Fig. 2e). Therefore, based on the similar results from two different measurements, we confirm the anomalous anisotropy of the spin current in Mn$_3$Pt which is not expected by the cubic crystal symmetry. The $\phi_E$ dependence of $H_{eff}^z/J$ can be well fitted using Eq. (3) (Fig. 3e), yielding parameters $\lambda_z = 3.48$ and $\mu_z = 9.59$. $\lambda_z/\mu_z = 0.36$ indicates that such an anomalous anisotropy is originated from the coexistence of $\mathscr{T}$-odd and $\mathscr{T}$-even SHE in Mn$_3$Pt, where $\mathscr{T}$-odd SHE is predominating. We have estimated the actual device temperature to be ~331 K for the maximum current in the AHE loop shift measurement (Supplementary Note 10), which is well below the $T_N$ of Mn$_3$Pt (~475 K). Therefore, the AFM spin texture and the associated spin current largely remain unaffected by the Joule heating effects.

## Field-free deterministic magnetization switching

The existence of $\sigma_{zx}^z$ allows the field-free deterministic switching of the perpendicular magnetization in the CoFeB layer adjacent to Mn$_3$Pt. However, such switching has not been achieved in Mn$_3$Pt by previous efforts. As will be shown below, our efforts in doing so not only end in success, but also provide solid evidence for the predominating role of the $\mathscr{T}$-odd SHE which, in turn, supports the mechanism proposed above for the anomalous anisotropy.

As shown in Fig. 4a, the injected pulse current, when above a threshold value, is able to change the sign of $R_{xy}$ for various $\phi_E$, indicating that deterministic magnetization switching of the PMA CoFeB layer has been realized without an in-plane assistant field (see "Methods" and Supplementary Note 11 for more details). The change of the switching Hall resistance is represented by $\triangle R_{xy}$, which is the half of the difference between $R_{xy}$ at zero current after the application of the positive and negative current pulses. We find that the $\phi_E$ dependence of $\triangle R_{xy}$ is consistent with that of $H_{eff}^z/J$ (Fig. 4b). The largest $\triangle R_{xy}$ corresponding to a switching ratio of $\approx77\%$ is achieved when $H_{eff}^z/J$ is maximum around $\phi_E = 45°$, indicating the optimal current direction for designing SOT devices based on Mn$_3$Pt. It is note that the actual device temperature is ~356 K for the maximum current in the magnetization switching measurement (Supplementary Note 10), and this temperature is well below the $T_N$ of Mn$_3$Pt.

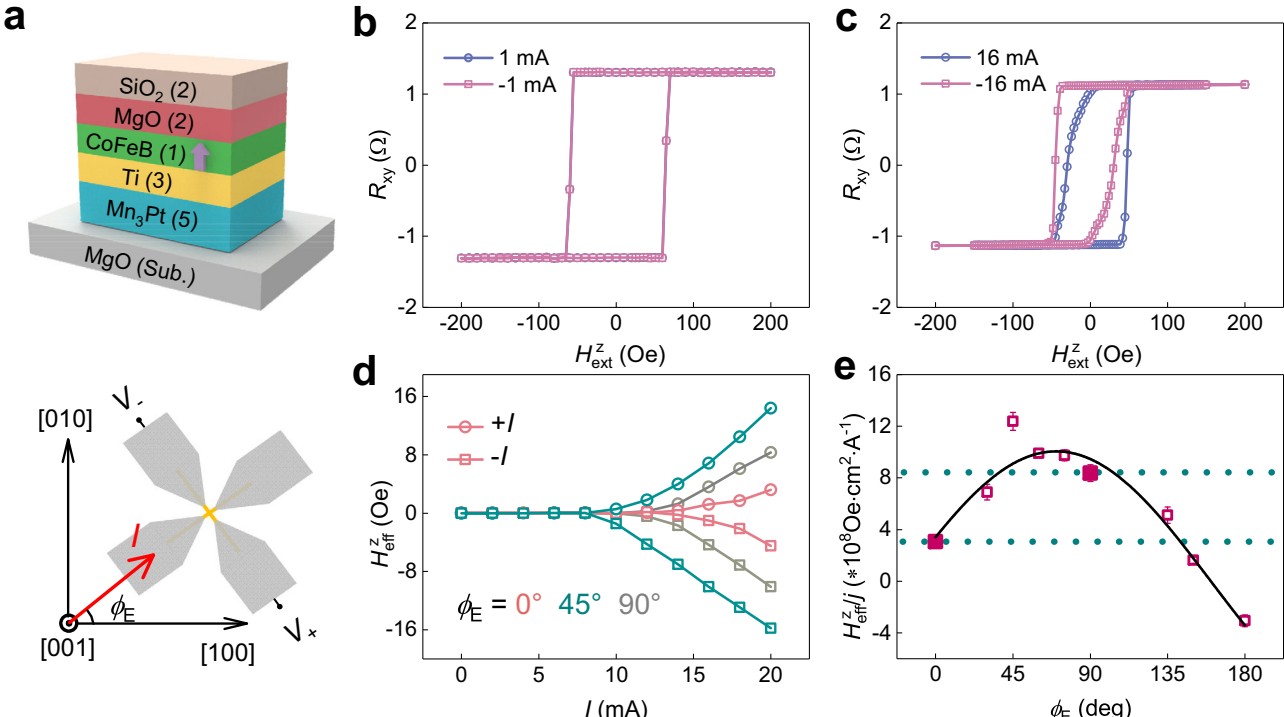

**Fig. 3 | AHE loop shift with a threshold current in Mn₃Pt/Ti/CoFeB/MgO/SiO₂.**
**a** (Upper) The schematic of the Mn₃Pt/Ti/CoFeB/MgO/SiO₂ heterostructure, with numbers in parentheses indicate layer thickness in nanometers. (Lower) The schematic of the AHE loop shift measurement setup. **b** The AHE loops under $I = \pm$ mA almost overlap with each other. **c** Under $I = \pm 1$ mA, the AHE loops show an obvious shift towards positive or negative values. **d** The $I$ dependence of $H_{\mathrm{eff}}^z$, which is defined as the shift of the AHE loop, at selected $\phi_E$ of 0°, 45°, and 90°. **e** The $\phi_E$ dependence of the $H_{\mathrm{eff}}^z$ per unit current density. The anisotropy between $\phi_E = 0°$ and $\phi_E = 90°$ is highlighted by the enlarged symbols and dashed horizontal lines. The solid line is a fit to $\lambda_z \cos\phi_E + \mu_z \sin\phi_E$. The error bars represent the standard deviations derived from measurements on three devices.

The magnetic space group $R\bar{3}m'$ supports a small but non-vanishing net magnetization in Mn₃Pt. The presence of such a net magnetization allows the switching of the magnetic order parameter by a small magnetic field, which is equivalent to the $\mathscr{T}$-operation which changes the sign of the $\mathscr{T}$-odd SHC but does not influence the $\mathscr{T}$-even SHC[29,46]. This property can be used to verify whether $\mathscr{T}$-odd SHE dominates the anisotropic spin current in Mn₃Pt and hence the field-free switching of the perpendicular magnetization. Here we perform the current-induced magnetization switching measurements for $\phi_E = 0°$ with the application of a premagnetization field $H_{\mathrm{pre}}$ of 8 T along the [001] direction, which aligns the magnetic order parameter of Mn₃Pt as depicted in the inset of Fig. 4c. The polarity of the field-free switching is anticlockwise in this case (Fig. 4d). Applying the $H_{\mathrm{pre}}$ to the [00$\bar{1}$] direction reverses the magnetic order parameter of Mn₃Pt and changes the switching polarity to clockwise. This clearly proves the predominating role of the $\mathscr{T}$-odd SHE[47] for the spin current in Mn₃Pt and hence the field-free switching of the perpendicular magnetization. We have performed additional AHE loop shift measurements following similar procedures to that adopted in the magnetization switching measurements (Supplementary Note 12). As shown in Fig. S12, the loop shifts for an $H_{\mathrm{pre}}$ of 8 T along the [001] and [00$\bar{1}$] directions are opposite, consistent with the reversal of the switching polarity.

## Discussion

We have demonstrated, via the measurements of ST-FMR, AHE loop shift, and current-induced magnetization switching, the existence of an anomalous anisotropy of the spin current generated by the spin source Mn₃Pt with a cubic crystal structure and a noncollinear anti-ferromagnetic order. While the observation of a spin polarization and the associated SOT along the out-of-plane direction able to induce field-free deterministic magnetization switching in a neighboring FM

layer is interesting in itself, the most significant finding of our work is the anisotropy of the SOT when the electrical current is injected along orthogonal cubic directions.

Unlike the AHE, the intrinsic contribution to the conventional $\mathscr{T}$-even SHE is expected to be isotropic in a nonmagnetic cubic spin source, i.e., it exhibits an isotropic SHC[7–9]. Anisotropic behavior of the conventional SHE in cubic spin sources has only been previously demonstrated in certain Pt-based heterostructures. For example, the spin Hall angle $\theta_{\mathrm{SH}}$ determined from the spin Hall magnetoresistance and the damping-like SOT were both found to be different between $I$ // [1$\bar{1}$0][42] or between $I$ // [1$\bar{1}$0] and $I$ // [11$\bar{2}$][38] in these het-erostructures. However, the two sets of orthogonal directions involved are not perfectly equivalent even from the crystallographic point of view, providing the opportunity for factors like the anisotropy in the resistivity $\rho$ and the spin diffusion length to impact the observed ani-sotropy via, e.g., $\theta_{\mathrm{SH}} \sim \rho \sigma_{ij}^{p,\mathrm{even}}$[48,49], although interfacial Rashba-Edelstein effect was also argued to contribute. In sharp contrast, the [100] and [010] directions of Mn₃Pt, along which we identify an ani-sotropy, are perfectly equivalent in the crystal structure. This equiv-alency is not affected even when the magnetic structure is taken into account. The $\mathscr{T}$-odd MSHE evidenced by the polarity reversal in the PMA magnetization switching justifies our proposal of the $\mathscr{T}$-odd MSHE and the $\mathscr{T}$-even conventional SHE combined to give rise to the anomalous anisotropy, a scenario that captures all the experimental observations, including specifically how the SOT efficiency extracted from ST-FMR, the out-of-plane effective field estimated from the AHE loop shift, and the current-induced magnetization switching ratio, vary with changing the injected current direction with respect to the crystal axes.

To our knowledge, our work presents the first observation−in a single material−of the coexistence of $\mathscr{T}$-even[12,13,50,51] and $\mathscr{T}$-odd

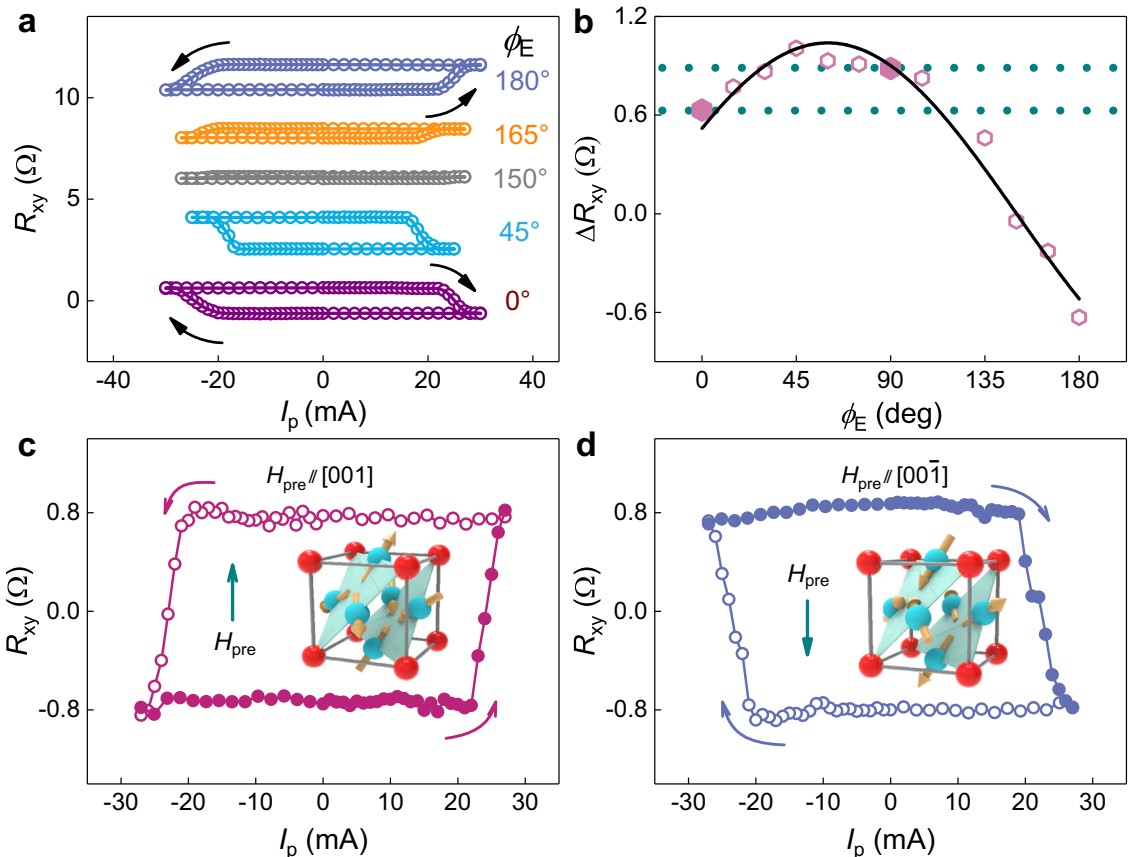

**Fig. 4 | Field-free deterministic magnetization switching and evidence for the MSHE in Mn₃Pt/Ti/CoFeB/MgO/SiO₂. a** The field-free deterministic switching of the CoFeB magnetization, represented by $R_{xy}$, for various $\phi_E$. **b** The $\phi_E$ dependence of the switching ratio, represented by $\triangle R_{xy}$, which is the difference between $R_{xy}$ at zero current after the application of the positive and negative current pulses. The anisotropy between $\phi_E = 0°$ and $\phi_E = 90°$ is highlighted by the enlarged symbols and dashed horizontal lines. The solid line is a fit to $\lambda_z \cos\phi_E + \mu_z \sin\phi_E$, suggesting again the combined effect of the conventional SHE and the MSHE. **c, d** The switching polarity is reversed with premagnetization fields along opposite directions, a hallmark of the presence of the MSHE. The insets illustrate the spin configuration.

SHE[24,29,39,40,47,52–58], which usually exist solely or independently in distinct material systems and do not show an intertwinement. The cooperation of the $\mathscr{T}$-even and $\mathscr{T}$-odd SHE offers anomalous transport anisotropy and useful functionality. The anomalous anisotropy uncovered here adds a new tuning knob to spintronic devices utilizing SOT: one could envision devices based on a cubic spin source, whose ability and efficiency in controlling the magnetization of the FM layer can be modified simply by changing the direction of the injected electrical current from one principle cubic axis to its orthogonal counterpart; meanwhile, most other properties are kept identical in this process due to the high cubic symmetry.

There are many other cubic kagome noncollinear antiferromagnets, such as Mn₃Ir[36], and Mn₃AN (A = Ga, Sn, Ni, etc.)[35,37,59] hosting a noncollinear magnetic order and a magnetic group symmetry similar to Mn₃Pt discussed in this work, which may also support the anisotropic spin currents. The strength of such anisotropy can be further enhanced by the exotic electronic structures in these materials. For example, the bulk Weyl cones and associated surface Fermi arcs have been observed in some kagome noncollinear antiferromagnets[60,61], which may strongly enhance the transport anisotropy. Moreover, in the cubic kagome noncollinear antiferromagnets, the strong correlation effect may occur[61,62], which may further enhance the transport anisotropy. We thus believe that cubic kagome noncollinear antiferromagnets are ideal material platforms for investigation of the anomalous transport anisotropy and realization of the efficient spintronic applications.

In summary, we observed an unexpected anisotropy of spin current in Mn₃Pt, a cubic structured spin source with noncollinear antiferromagnetism due to the intertwined $\mathscr{T}$-odd and $\mathscr{T}$-even SHE. We also show the spin current generated in Mn₃Pt can be used to realize efficient field-free SOT switching of ferromagnets PMA and the anomalous anisotropy can be used to optimize the switching performance. Our work offers a new route to introduce transport anisotropy in materials with high crystal symmetry, which is beneficial for designing and engineering of low-power and high-performance electronic devices.

## Methods
### Sample preparation
Samples of Mn₃Pt, Mn₃Pt/Ti/CoFeB/MgO/SiO₂, Mn₃Pt/Py, and Pt/Py bilayers were deposited on MgO(001) substrates by DC/RF magnetron sputtering with a base pressure of $1 \times 10^{-7}$ Torr. Note that the metallic and oxide films were deposited by using DC and RF magnetron sputtering, respectively. For the deposition of Mn₃Pt, the MgO(001) substrate was pre-annealed for 1 h at 700 °C before deposition to obtain a smooth and clean substrate. The deposition was performed at 500 °C. The Ar gas pressure and sputtering power were 2 mTorr and 40 W, respectively. After deposition, the Mn₃Pt film was heated to 550 °C for 1.5 h to improve the crystalline quality. After cooling down to room temperature, the Ti/CoFeB/MgO/SiO₂ multilayer or the Py layer were deposited onto the Mn₃Pt film, respectively. The Mn₃Pt/Ti/CoFeB/MgO/SiO₂ heterostructure was in situ annealed at 200 °C for 30 min under vacuum conditions to promote PMA. A 2 nm SiO₂ capping layer was used to protect its underlayers. For the Mn₃Pt(12 nm)/Py (Ni₈₀Fe₂₀,

13 nm) bilayer used in the ST-FMR measurements, the Py film was prepared at room temperature, with the Ar gas pressure and sputtering power being 2 mTorr and 40 W, respectively. The control sample Pt/Py bilayers used for the ST-FMR measurements were deposited at room temperature and the Ar gas pressure and sputtering power for the Pt film were 2 mTorr and 20 W, respectively. The film thicknesses were controlled by the deposition time with a pre-calibrated deposition rate as determined by X-ray reflectivity measurements.

### Device fabrication

In order to investigate the anisotropy of the spin-orbit torque of cubic $Mn_3Pt$, samples of $Mn_3Pt$/Ti/CoFeB/MgO and $Mn_3Pt$ (Pt)/Py were patterned into Hall bars ($10\,\mu m \times 50\,\mu m$) and microstrip devices ($20\,\mu m \times 50\,\mu m$), respectively, using a combination of photolithography and ion beam etching. Then, a top electrode of Ti(5 nm)/Cu(100 nm) was deposited by DC magnetron sputtering. For devices with different current directions in the sample plane, $\phi_E$ ranges from 0° to 180° with a step of 15°.

### Sample characterization

The thickness and crystal structure were characterized by X-ray reflectivity and high-resolution X-ray diffraction techniques with a Bruker D8 Discover diffractometer using Cu $K_\alpha$ radiation ($\lambda = 0.15419$ nm). The cross-sectional crystalline structure was imaged by AC-STEM (FEI Titan Themis 200) operated at 200 kV. The atomic ratio of our sample has been checked by energy-dispersive X-ray spectroscopy (EDS). The magnetic and electrical properties were measured in a magnetic property measurement system (MPMS, Quantum Design) and physical property measurement system (PPMS, Quantum Design), respectively.

### ST-FMR measurements

The ST-FMR signals ($V_{mix}$) were measured by a Stanford Research SR830 lock-in amplifier. In the angular-dependent ST-FMR measurements, the applied microwave current with frequency and nominal power were 7 GHz and 18 dBm, respectively.

### AHE loop shift measurements

The existence of $H_{eff}^z$ was verified by the AHE loop shift measurements with different pulse currents, where $H_{eff}^z$ is defined as the shift of the loop $H_{eff}^z(I) = [|H_{rev}^+(I)| - |H_{rev}^-(I)|]/2$, with $H_{rev}^\pm(I)$ being the positive and negative magnetization-reversal fields.

### Current-induced magnetization switching measurements

The current-induced magnetization measurements were conducted by utilizing a Keithley 6221 current source and a 2182 nano voltmeter. For each experimental data point in the $R_{xy}$-$I$ loop, a pulse d.c. current $I_p$ with a duration of 200 μs was applied to the Hall bar device as the write current. Then, a small probe pulse current of 0.1 mA with a duration of 2 ms was applied to monitor the $R_{xy}$. The amplitude of the write pulse current was varied to obtain a complete $R_{xy}$-$I$ loop.

## Data availability

The data that support the findings of this study are available from the corresponding authors upon reasonable request.

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

## Acknowledgements

This work was supported by the National Key Research and Development Program of China (Grant Nos. 2021YFA1600200), the National Natural Science Foundation of China (Grant Nos. 12174103, 12274411, 12274125, 12241405, 12274411 and 52250418), the Natural Science Foundation of Shanghai (Grant Nos. 21ZR1420500 and 21JC1402300), the Shanghai Pujiang Program (Grant No. 21PJ1403100), the CAS Project for Young Scientists in Basic Research (Grant No. YSBR-084), and the Natural Science Foundation of Anhui Province (Grant No. 2208085QA08).

## Author contributions

C.C. and S.C. conceived the idea and designed the experiment. D.F.S., Y.X., J.C., and Q.Z. supervised the project. C.C. and Y.W. grew the samples and performed the structural characterizations. S.C. fabricated the devices. C.C. and S.C. performed electrical transport measurements and analyzed the results together with X.Q. Z.Z. and G.Y. performed the ST-FMR measurements. L.L., T.Z., and J.C. gave suggestions on the experiments. R.C.X. and D.F.S. performed the theoretical analyses. All authors contributed to discussions. C.C., S.C., D.F.S., and Y.X. wrote the manuscript with input from all authors.

## Competing interests

The authors declare no competing interests.
