## [Peer Review File · Nature Communications]

Reviewers' Comments:

Reviewer #1:

Remarks to the Author:

In the work entitled "Anomalous anisotropy of spin current in a cubic spin source with noncollinear antiferromagnetism", Cao et al. reports an observation of anomalous spin current anisotropy in a noncollinear antiferromagnet, which appears to be highly unconventional. In particular, the authors have demonstrated, via the measurements of ST-FMR, AHE loop shift, and current induced magnetization switching, the existence of an anomalous anisotropy of the spin current generated by the spin source Mn₃Pt with a cubic crystal structure and a noncollinear antiferromagnetic order. Note that Mn₃Pt also host a kagome lattice, which is geometrical nontrivial and often thought to be frustrated. The explorations of novel transport behaviors for noncollinear antiferromagnets and kagome materials are emerging at the frontier of condensed matter physics. I evaluate this device work to be highly systematic and the novel observation certainly advances this research frontier. I can recommend the publication of this work if the authors can further address the comments below:

1. I feel the title is a little bit lengthy. I suggest to make it more concise, for instance: "Anomalous spin current anisotropy in a noncollinear antiferromagnet". A concise title may eventually help to promote the impact of this work.
2. From the electronic structure point of view, the Mn₃X type of kagome materials has long been proposed as Weyl magnets (see , Nature 612, 647–657 (2022) for example). In this Weyl picture, it can be seen that both the bulk Weyl cones distribution and surface Fermi arc distributions break rotational symmetry and are highly anisotropic. I would like to see some relevant discussions between this electronic anisotropy and the authors' observations.
3. Mn₃X materials have also been reported to be correlated (Nat. Mater. 16, 1090–1095 (2017), Phys. Rev. Lett. 125, 046401 (2020)). Correlation can often driven electronic nematicity that leading to strong anisotropy. This correlation aspects also need to be discussed.

Reviewer #2:

Remarks to the Author:

In this work by Cao and co-workers, it is reported that an anomalous anisotropy of the spin current is observed by using a cubic system Mn₃Pt. T_{even} and T_{odd} spin Hall effects are found to be coexist in such non-collinear antiferromagnetic material system, which was examined by angle-dependent spin-torque FMR in samples with in-plane anisotropy and the so-called loop shift measurement in samples with PMA. Finally, field-free current-induced switching measurement was performed to showcase the usefulness of the unconventional z-spin. Although the data presentation is well-organized, there are some fundamental issues in this research that prevent me from recommending this work to be further considered as an article in Nature Communications:

1. Mn₃X system has been studied widely for the past few years, including using Mn₃Sn for field-free switching by some of the co-authors in this work. This work serves as more an extension to previous studies.
2. Moreover, these materials need to be grown on single crystal substrate, such as MgO(001) in this case, to generate the unconventional spin currents. It is of rare chance that these materials can be employed by industry to demonstrate robust field-free switching.
3. It is also important for the authors to show, for example, HR-TEM results to verify that the physical phenomenon that they claimed are indeed related to the crystal structure and the antiferromagnetic ordering. Otherwise, the observed effect might also be originated from other mechanisms such as tilted magnetic anisotropy. Also, the atomic ratio of Mn₃Pt₁ also needs to be verified, since the sputtered film might deviate from the target in terms of stoichiometry.

4. Since the field-free switching behavior depends on the nonvanishing net magnetization in Mn₃Pt, the ST-FMR and loop shift measurement results should also show such dependency. The authors need to verify this.

5. What is the spin diffusion length of Ti in the present work? It is surprising that the unconventional spin current can diffuse through 3nm of Ti and still be efficient enough to drive hysteresis loop shift and current-induced switching.

6. What is the estimated temperature of the device during switching measurements? A 200 us wide pulse is not a short one, and might generate significant heating to the device. Also, please provide the resistivity characterization result for Mn₃Pt.

Response to Reviewers

We deeply appreciate the valuable comments and helpful suggestions by both Reviewers. Here we respond to the comments in detail. Revisions are highlighted in red color in the revised manuscript.

Reviewer #1:

General Comment: In the work entitled "Anomalous anisotropy of spin current in a cubic spin source with noncollinear antiferromagnetism", Cao *et al.* report an observation of anomalous spin current anisotropy in a noncollinear antiferromagnet, which appears to be highly unconventional. In particular, the authors have demonstrated, via the measurements of ST-FMR, AHE loop shift, and current induced magnetization switching, the existence of an anomalous anisotropy of the spin current generated by the spin source Mn_3Pt with a cubic crystal structure and a noncollinear antiferromagnetic order. Note that Mn_3Pt also hosts a kagome lattice, which is geometrically nontrivial and often thought to be frustrated. The explorations of novel transport behaviors for noncollinear antiferromagnets and kagome materials are emerging at the frontier of condensed matter physics. I evaluate this device work to be highly systematic and the novel observation certainly advances this research frontier. I can recommend the publication of this work if the authors can further address the comments below:

We thank the Reviewer for reading our manuscript and for his/her valuable comments. We are pleased that the Reviewer finds that our work “to be highly systematic and the novel observation certainly advances this research frontier”.

Q1. I feel the title is a little bit lengthy. I suggest making it more concise, for instance: Anomalous spin current anisotropy in a noncollinear antiferromagnet". A concise title may eventually help to promote the impact of this work.

We appreciate this suggestion by the Reviewer. The title has been modified accordingly.

Q2. From the electronic structure point of view, the Mn_3X type of kagome materials has long been

proposed as Weyl magnets [see, Nature 612, 647–657 (2022) for example]. In this Weyl picture, it can be seen that both the bulk Weyl cones distribution and surface Fermi arc distributions break rotational symmetry and are highly anisotropic. I would like to see some relevant discussions between this electronic anisotropy and the authors' observations.

We thank the Reviewer for this comment. Indeed, the magnetic point group symmetry only offers the possibility of the transport anisotropy, while its strength should be influenced by many other electronic factors. Although the Weyl points have not been observed in Mn_3Pt according to previous theoretical and experimental reports, it may emerge in other cubic kagome noncollinear antiferromagnets and strongly influence the anisotropy. We thus add a discussion of this aspect in third paragraph on page 11 according to the Reviewer's suggestion, as highlighted in red in the revised manuscript. The reference the reviewer mentioned has now been cited as Ref. 60 in this revision.

We note in passing that the Mn_3X type kagome materials in which Weyl points have been considered are of *hexagonal* type, including Mn_3Sn and Mn_3Ge , while Mn_3Pt is of *cubic* type. To highlight this difference and avoid ambiguity, we now change the notation of “ Mn_3X ($X= Pt, Ir, or Rh$)” to “ Mn_3Y ($Y= Pt, Ir, or Rh$)” in the revised manuscript.

3. Mn_3X materials have also been reported to be correlated [Nat. Mater. 16, 1090–1095 (2017), Phys. Rev. Lett. 125, 046401 (2020)]. Correlation can often drive electronic nematicity that leads to strong anisotropy. These correlation aspects also need to be discussed.

We thank the Reviewer for this comment. The Reviewer is correct that the correlation effect may strongly enhance the transport anisotropy in materials with the same symmetry as Mn_3Pt . We have discussed this possibility in third paragraph on page 11 according to the Reviewer's suggestion, as highlighted in red in the revised manuscript. The two papers the reviewer suggested have now been cited as Refs. 61 and 62 in this revision.

Reviewer #2:

General Comment: In this work by Cao and co-workers, it is reported that an anomalous anisotropy of the spin current is observed by using a cubic system Mn_3Pt . T -even and T -odd spin Hall effects are found to coexist in such non-collinear antiferromagnetic material system, which was examined by angle-dependent spin-torque FMR in samples with in-plane anisotropy and the so-called loop shift measurement in samples with PMA. Finally, field-free current-induced switching measurement was performed to showcase the usefulness of the unconventional z -spin. Although the data presentation is well-organized, there are some fundamental issues in this research that prevent me from recommending this work to be further considered as an article in Nature Communications.

We thank the Reviewer for reading our manuscript and for his/her valuable comments. We are pleased that the Reviewer finds that in our work the “data presentation is well-organized”. We articulate our opinion in the point-to-point response to the reviewer’s comments. We do hope that our response and additional experiments and simulations would encourage the reviewer to be more positive regarding the impact of our work.

Q1. Mn_3X system has been studied widely for the past few years, including using Mn_3Sn for field-free switching by some of the co-authors in this work. This work serves as more of an extension of previous studies.

We respectfully disagree with the Reviewer for this comment. It is indeed true that both hexagonal Mn_3X ($X= Sn, Ge, \text{ or } Ga$) and cubic Mn_3Y ($Y= Pt, Ir, \text{ or } Rh$) systems (please see the response to comment No. 2 of Reviewer #1 for more information about the change of notation) have attracted great attention in recent years. However, we emphasize that the novelty of this work, as articulated in the introduction part, is the demonstration of an unexpected anisotropy in material systems with high crystal symmetry. The clarification of its exotic nature not only offers a conceptual breakthrough in the *fundamental* understanding of spin-dependent transport properties in quantum materials, but also helps the realization of high-performance spintronics. Here, we would like to emphasize the novelty and importance of this work in detail in the following arguments:

- 1). **Anomalous transport anisotropy in a high-symmetry system.** A high-symmetry cubic material has been widely considered to be isotropic for transport properties. In contrast to this common expectation, we show, for the first time, that an anomalous transport anisotropy can emerge in a high-symmetry (001) film of a cubic material, due to the remodulation of the electronic structures by the noncollinear antiferromagnetism. This unexpected anomalous transport anisotropy is expected to exist widely in cubic kagome antiferromagnets, including but not limit to Mn₃Pt.
- 2). **Intertwinement of \mathcal{T} -even and \mathcal{T} -odd SHE.** For the first time, we observe the coexistence of \mathcal{T} -even and \mathcal{T} -odd SHE, and their cooperation offers anomalous transport anisotropy and useful functionality. The \mathcal{T} -even and \mathcal{T} -odd SHE usually exist solely or independently in material systems and do not show such an intertwinement.
- 3). **Promising functionalities driven by the anisotropic spin current.** The observed anisotropic spin current results in useful functionalities accelerating spintronic applications. As one typical example, we show that the SOT generated by the intertwined \mathcal{T} -even and \mathcal{T} -odd SHE in Mn₃Pt can switch a perpendicular-magnetized ferromagnet without the external magnetic field, showing the great advantage compared to that resulting from the conventional SHE. Benefiting from the anomalous anisotropy of the spin current, the efficiency of SOT can be further optimized for low-power spintronic applications.

We hope therefore that the Reviewer would agree with our arguments and reconsider his/her negative opinion on our work.

Q2. Moreover, these materials need to be grown on single crystal substrates, such as MgO (001) in this case, to generate the unconventional spin currents. It is of rare chance that these materials can be employed by industry to demonstrate robust field-free switching.

We respectfully disagree with the Reviewer for this comment. First, we want to emphasize again that the novelty of our work lies mainly in the fundamental importance, i.e., the demonstration of the anisotropic spin current, which is an unexpected transport anisotropy of high-symmetry cubic systems, as explained in detail in the response to comment 1. This property is intrinsic for cubic kagome antiferromagnets such as Mn₃Pt. Although it would be more pronounced in single-crystalline bulk or thin film samples, its existence should not rely on the form of the sample or substrates chosen. Second,

we want to point out that although we used MgO as the substrate in this work, the epitaxy growth of cubic kagome antiferromagnets such as Mn₃Pt can be grown epitaxially both on a cubic substrate or cubic buffer layer. A typical example is the recently reported Mn₃Pt/MgO/Mn₃Pt antiferromagnetic tunnel junction, where the bottom Mn₃Pt layer is grown on a MnPt layer, and the top Mn₃Pt layer is grown on a MgO layer [See Qin et al., Nature 613, 485 (2023)]. We thus believe that the demonstration of the anisotropic spin current is not a problem for the industry application of the cubic kagome antiferromagnets such as Mn₃Pt, as a cubic buffer layer is convenient to achieve using modern thin film growth techniques.

Q3. It is also important for the authors to show, for example, HR-TEM results to verify that the physical phenomena that they claimed are indeed related to the crystal structure and the antiferromagnetic ordering. Otherwise, the observed effect might also be originated from other mechanisms such as tilted magnetic anisotropy. Also, the atomic ratio of Mn₃Pt₁ also needs to be verified, since the sputtered film might deviate from the target in terms of stoichiometry.

We appreciate the suggestion by the Reviewer to add more details about the basic characterizations of the sample. The cross-sectional crystalline structure of the Mn₃Pt layer on the MgO(001) substrate imaged by aberration-corrected scanning transmission electron microscopy (AC-STEM, FEI Titan Themis 200) operated at 200 kV is shown in Fig. R1, evidencing a good epitaxial growth of cubic Mn₃Pt on cubic MgO with sharp interfaces. The atomic ratio of our sample has been checked by energy-dispersive X-ray spectroscopy (EDS). As shown in Fig. R2, the atomic ratio is determined to be Mn : Pt = 74.69 : 25.31 \approx 3 : 1. We have also confirmed the magnetic hysteresis and anomalous Hall effects of the Mn₃Pt film, as shown in Fig. S1 in the initial submission. These properties are consistent with previous reports of Mn₃Pt, indicating the crystal and magnetic structure shown in Fig. 1c is well sustained in the thin film we prepared. Moreover, we have found that the anisotropy of the spin current can be well fitted using Eq. (3) in the manuscript, and the switching polarity can be reversed by a premagnetization field. These phenomena cannot be explained by mechanisms such as a tilted magnetic anisotropy, but can be well understood by the intertwinement of the \mathcal{T} -even and \mathcal{T} -odd spin Hall effects in Mn₃Pt.

In this revision, we have added the TEM image and the EDS results in Note 2 in the Supplementary Information. The Figs. R1 and R2 are added as Figs. S2a and S2b.

Figure R1 The AC-STEM image of the interfacial region of the MgO(001)/Mn₃Pt heterostructure.

Figure R2 The atomic ratio of Mn₃Pt measured by EDS.

Q4. Since the field-free switching behavior depends on the nonvanishing net magnetization in Mn_3Pt , the ST-FMR and loop shift measurement results should also show such dependency. The authors need to verify this.

We thank the reviewer for this comment. As mentioned in the manuscript, the small but nonvanishing net magnetization is allowed by the $R\bar{3}m'$ magnetic space group of Mn_3Pt , which provides a tuning knob for the switching of the magnetic order parameter and the associated reversal of the \mathcal{T} -odd spin current in Mn_3Pt by a magnetic field. The latter is responsible for the different switching polarities under the opposite premagnetization field shown in Fig. 4. Following the Reviewer's suggestion, we have performed additional AHE loop shift measurements following similar procedures to that adopted in the magnetization switching measurements. As shown in Fig. R3, the loop shifts for an H_{pre} of 8 T along the $[001]$ and $[00\bar{1}]$ directions are opposite, consistent with the reversal of the switching polarity we have shown in the manuscript.

In this revision, we have added the relevant information and discussions in third paragraph on page 9 of the revised manuscript, and Note 12 in the Supplementary Information. The Fig. R3 has been added as Fig. S12 in the Supplementary Information.

Figure R3 a-c The AHE loops under $I = \pm 1$ mA, ± 20 mA, and the I dependence of H_z^{eff} at selected $\phi_E = 0^\circ$ with a premagnetization field of 8 T along the $[001]$ direction. d-f Similar to (a-c) but with a premagnetization field of 8 T along

the $[00\bar{1}]$ direction.

Q5. What is the spin diffusion length of Ti in the present work? It is surprising that the unconventional spin current can diffuse through 3nm of Ti and still be efficient enough to drive hysteresis loop shift and current-induced switching.

We thank the Reviewer for this comment. The spin diffusion length in Ti has been reported to be $\lambda_s^{\text{Ti}} \sim 13.3$ nm [see Du et al., Phys. Rev. B 90, 140407(R) (2014)], which is much larger than the thickness of Ti layer in our SOT device. Also, in our previous work [ACS Nano 16, 12727 (2022)], we find a 3 or 6 nm Ti layer does not significantly attenuate the spin current. Therefore, we believe the Ti layer does not influence the main conclusion of this work.

Q6. What is the estimated temperature of the device during switching measurements? A 200 us wide pulse is not a short one, and might generate significant heating to the device. Also, please provide the resistivity characterization result for Mn₃Pt.

We appreciate the suggestion by the Reviewer. We have added more details about the basic characterizations of the sample, as described below:

To determine the actual temperature of the device during the loop shift and magnetization switching measurements, we first measured the temperature dependence of the longitudinal resistance R_{xx} under a small DC current $I \sim 0.1$ mA. The actual temperature T of the device can be estimated by the change of R_{xx} based on the R_{xx} versus T curve in Fig. R4a.

Since the resistance R_{xx} of our Mn₃Pt(5)/Ti(3)/CoFeB(1)/MgO(2)/SiO₂(2) device is in the range of k Ω , the measurement range of the voltage meter needs to be adjusted to 100 V during the measurement, and the pulse width is required to be larger than 600 μ s in the resistance measurements. In contrast, we use a pulse width of 200 μ s in the loop shift and magnetization switching measurements. To compare the results in different measurements with different pulse widths, we plot R_{xx} as a function of the pulse width in Fig. R4b. The data is shown to follow the empirical formula $R_{xx} = \frac{\beta}{\alpha - I^2 \times t}$, where I and t are the pulse current magnitude and pulse width, respectively, and α and β are constant fitting parameters

[See Liu et al., Phys. Rev. B 101, 220402(R) (2020)]. By extrapolating the linear fit of $1/R_{xx}$ to a pulse width of 200 μs , we can obtain the resistance R_{xx} with 200 μs pulse width. We plot R_{xx} versus I in Fig. R4c and estimate the actual temperature with each pulse current by converting R_{xx} into actual temperature as shown in Fig. R4d. Based on these measurements, we estimate the actual device temperature to be ~ 331 K for the maximum current in the AHE loop shift measurement (Fig. 3 of the manuscript) and ~ 356 K for the maximum current in the magnetization switching measurement (Figs. 4c and 4d of the manuscript), which are well below the T_N of Mn_3Pt (~ 475 K).

In this revision, we have added relevant information and discussions in second paragraph on page 8 and second paragraph on page 9 of the revised manuscript, and Note 10 of the Supplementary Information. The Fig. R4 has been added as Fig. S10 in the Supplementary Information.

Figure R4 a The resistance R_{xx} of the $\text{Mn}_3\text{Pt}(5)/\text{Ti}(3)/\text{CoFeB}(1)/\text{MgO}(2)/\text{SiO}_2(2)$ device as a function of the temperature. The red line is a linear fit. **b** The pulse width dependence of the conductance of the $\text{Mn}_3\text{Pt}(5)/\text{Ti}(3)/\text{CoFeB}(1)/\text{MgO}(2)/\text{SiO}_2(2)$ device. The red line is a linear fit. **c** The resistance of the $\text{Mn}_3\text{Pt}(5)/\text{Ti}(3)/\text{CoFeB}(1)/\text{MgO}(2)/\text{SiO}_2(2)$ device versus the pulse current amplitude. (Current pulse width = 200 μs). **d** The actual device temperature with different pulse amplitudes. (Current pulse width = 200 μs).

Reviewers' Comments:

Reviewer #1:

Remarks to the Author:

I thank the authors for addressing my comments, and can recommend the publication of this work.

Reviewer #2:

Remarks to the Author:

I would like to thank the authors for addressing my concerns carefully. The authors have shown additional experimental data, such as HR-TEM images and AHE loop shift measurement results, to respond to my technical questions. However, I'm still doubtful regarding the claim of the work's novelty and especially the possibility of realistic device integration with the existing CMOS technology. Note that since the claimed mechanism here relies on epitaxial growth of cubic kagome antiferromagnet, "wafer-scale" single crystalline Mn₃Pt deposition is required. This is unlikely to be feasible in, for example, back-end-of-line MRAM processing.

Both Reviewers expressed their appreciation for the revisions we made. Reviewer #1 has made an explicit recommendation for publication. The only concern from Reviewer #2 now is about the industrial application of the mechanism we uncovered. We respond to this below:

Reviewer #2: “I would like to thank the authors for addressing my concerns carefully. The authors have shown additional experimental data, such as HR-TEM images and AHE loop shift measurement results, to respond to my technical questions. However, I'm still doubtful regarding the claim of the work's novelty and especially the possibility of realistic device integration with the existing CMOS technology. Note that since the claimed mechanism here relies on epitaxial growth of cubic kagome antiferromagnet, "wafer-scale" single crystalline Mn₃Pt deposition is required. This is unlikely to be feasible in, for example, back-end-of-line MRAM processing.”

In terms of Reviewer #2's concern about the realistic device integration with the existing CMOS technology, we argue that wafer-scale single-crystalline Mn₃Pt deposition is technically possible. We agree with Reviewer #2 that the device application of Mn₃Pt indeed relies on more future works beyond the present one.

In our work, we use magnetron sputtering on single-crystalline MgO substrates for the demonstration of the anomalous spin current anisotropy. However, as we argued in our previous response letter, the epitaxy growth of cubic kagome antiferromagnets such as Mn₃Pt does not necessarily requires a single-crystalline cubic substrate. The cubic kagome antiferromagnets can

also be epitaxially grown on a cubic buffer layer, which is convenient to obtain in the industry. Even for devices that adopt a single-crystalline cubic substrate, the successful deposition of wafer-scale single-crystalline Mn_3Pt boils down to the successful growth of wafer-scale single-crystalline MgO . In fact, single-crystalline MgO wafers with a size of up to ~ 2 inches are readily available in the industry.

More importantly, as emphasized in our previous response letter, the novelty of our work lies mainly in the *fundamental/conceptual* aspect. Large-scale industrial applications may indeed call for more future efforts, which is beyond the scope of our work. This, however, should not degrade the significance of our *proof-of-concept* work. In fact, the ideas underlying various types of electronic/spintronic devices with practical applications were historically first demonstrated in prototypical systems containing single crystals. As an example, the giant tunneling magnetoresistance in magnetic tunnel junctions—building blocks of MRAM—was first demonstrated in fully epitaxial $\text{Fe}/\text{MgO}/\text{Fe}$ multilayers with a single-crystalline MgO (001) barrier. The ensuing industrial application, in contrast, adopts a $\text{CoFeB}/\text{MgO}/\text{CoFeB}$ structure, where (001)-oriented polycrystalline MgO can be grown on amorphous CoFeB .

We believe that with this clarification, our manuscript is now ready for the production process.